# Constraint-based Causal Structure Learning with Consistent Separating Sets

**Honghao Li,   Vincent Cabeli,   Nadir Sella,   Hervé Isambert**[*]
Institut Curie, PSL Research University, CNRS UMR168, Paris
{honghao.li, vincent.cabeli, nadir.sella, herve.isambert}@curie.fr

## Abstract

We consider constraint-based methods for causal structure learning, such as the PC algorithm or any PC-derived algorithms whose first step consists in pruning a complete graph to obtain an undirected graph skeleton, which is subsequently oriented. All constraint-based methods perform this first step of removing dispensable edges, iteratively, whenever a separating set and corresponding conditional independence can be found. Yet, constraint-based methods lack robustness over sampling noise and are prone to uncover spurious conditional independences in finite datasets. In particular, there is no guarantee that the separating sets identified during the iterative pruning step remain consistent with the final graph. In this paper, we propose a simple modification of PC and PC-derived algorithms so as to ensure that all separating sets identified to remove dispensable edges are consistent with the final graph, thus enhancing the explainability of constraint-based methods. It is achieved by repeating the constraint-based causal structure learning scheme, iteratively, while searching for separating sets that are consistent with the graph obtained at the previous iteration. Ensuring the consistency of separating sets can be done at a limited complexity cost, through the use of block-cut tree decomposition of graph skeletons, and is found to increase their validity in terms of actual d-separation. It also significantly improves the sensitivity of constraint-based methods while retaining good overall structure learning performance. Finally and foremost, ensuring sepset consistency improves the interpretability of constraint-based models for real-life applications.

## 1   Introduction

While the oracle versions of constraint-based methods have been demonstrated to be sound and complete (Zhang, 2008; Spirtes, Glymour, and Scheines, 2000; Pearl, 2009), a major limitation of these methods is their lack of robustness with respect to sampling noise for finite datasets. This has largely limited their use to analyze real-life data so far, although important advances have been made lately, in particular, to limit the order-dependency of constraint-based methods (Colombo and Maathuis, 2014) or to improve their robustness to sampling noise by recasting them within a maximum likelihood framework (Affeldt and Isambert, 2015; Affeldt, Verny, and Isambert, 2016).

However, it remains that constraint-based methods still lack graph consistency, in practice, as they do not guarantee that the learnt structures belong to their presumed class of graphical models, such as a completed partially directed acyclic graph (CPDAG) model for the PC (Spirtes and Glymour, 1991; Kalisch and Bühlmann, 2008; Kalisch et al., 2012) or IC (Pearl and Verma, 1991) algorithms, or a partial ancestral graph (PAG) for FCI or related constraint-based algorithms allowing for unobserved latent variables (Spirtes, Meek, and Richardson, 1999; Richardson and Spirtes, 2002; Colombo et al., 2012; Verny et al., 2017; Sella et al., 2018). By contrast, search-and-score structure learning

---

[*]corresponding author

methods (Koller and Friedman, 2009) inherently enforce graph consistency by searching structures within the assumed class of graphs, *e.g.*, within the class of directed acyclic graphs (DAG). Similarly, hybrid methods such as MMHC (Tsamardinos, Brown, and Aliferis, 2006) can also ensure graph class consistency by maximizing the likelihood of edge orientation within the class of DAGs.

This paper concerns, more specifically, the inconsistency of separating sets used to remove dispensable edges, iteratively, based on conditional independence tests. This inconsistency arises as some separating sets might no longer be compatible with the final graph, if they were not already incompatible with the current skeleton, when testing for conditional independence during the pruning process. It occurs, for instance, when a node in a separating set is not on any indirect path linking the extremities of a removed edge, as noted in (Spirtes, Glymour, and Scheines, 2000). Such inconsistencies can be seen as a major shortcoming of constraint-based methods, as the primary motivation to learn and visualize graphical models is arguably to be able to read off conditional independences directly from the graph structure (Spirtes, Glymour, and Scheines, 2000; Pearl, 2009).

In the following, we propose a simple modification of PC or PC-derived algorithms so as to ensure that all conditional independences identified and used to remove dispensable edges are consistent with the final graph. It is achieved by repeating the constraint-based causal structure learning scheme, iteratively, while searching for separating sets that are consistent with the graph obtained at the previous iteration, until a limit cycle of successive graphs is reached. The union of the graphs over this limit cycle is then guaranteed to be consistent with the separating sets and corresponding conditional independences used to remove all dispensable edges from the initial complete graph. Enforcing sepset consistency of constraint-based methods is found to limit their tendency to uncover spurious conditional independences early on in the pruning process when the combinatorial space of possible separating sets is still large. As a result, enforcing sepset consistency reduces the large number of false negative edges usually predicted by constraint-based methods (Colombo and Maathuis, 2014) and, thereby, achieve a better balance between their sensitivity and precision. Ensuring the consistency of separating sets is also found to increase their validity in terms of actual d-separation and, therefore, to improve the interpretability of constraint-based models for real-life applications. Moreover, ensuring the consistency of separating sets can be done at a limited complexity cost, through the use of block-cut tree decomposition of graph skeletons, which enables to learn causal structures with consistent separating sets for a few hundred nodes. By contrast, earlier methods aiming at reducing the number of d-separation conflicts or other structural inconsistencies through SAT-based approaches, *e.g.* (Hyttinen et al., 2013), have a much larger complexity burden, which limits their applications to very small networks in practice.

## 2 Result

### 2.1 Background

#### 2.1.1 Terminology

A **graph** $\mathcal{G}(V, E)$ consists of a **vertex set** $V = \{X_1, \cdots, X_p\}$ and an **edge set** $E$. All graphs considered here have at most one edge between any pair of vertices. A **walk** is a sequence of edges joining a sequence of vertices. A **trail** is a walk without repeated edge. A **path** is a trail without repeated vertices. A **cycle** is a trail in which the only repeated vertices are the first and last vertices. Vertices are said to be **adjacent** if there is an edge between them. If all pairs of vertices in a graph are adjacent, it is called a **complete graph** and is denoted by $\mathcal{G}_c$. By constrast, an **empty graph**, denoted by $\mathcal{G}_\emptyset$, consists of isolated vertices with no edges. The **adjacency set** of a vertex $X_i$ in a graph $\mathcal{G}$, denoted by $\mathrm{adj}(\mathcal{G}, X_i)$, is the set of all vertices in $V$ that are adjacent to $X_i$ in $\mathcal{G}$. If an edge is directed, as $X \to Y$, $X$ is a parent of $Y$ and $Y$ a child of $X$. A **collider** is a triple $(X_i, X_j, X_k)$ in a graph where the edges are oriented as $X_i \to X_k \leftarrow X_j$. A **v-structure** is a **collider** for which $X_i$ and $X_j$ are not adjacent. Given a statistical significance level $\alpha$, the **conditional independence** of a pair of variables $(X_i, X_j)$ given a set of variables $C$, is denoted by $(X_i \perp\!\!\!\perp X_j | C)_\alpha$, where $C$ is called a **separating set** or "**sepset**" for $(X_i, X_j)$.

#### 2.1.2 The PC and PC-stable Algorithms

The PC algorithm (Spirtes and Glymour, 1991), outlined in algorithm 1, is the archetype of constraint-based structure learning methods (Spirtes, Glymour, and Scheines, 2000; Pearl, 2009), as illustrated

in Figure 1. Given a dataset over a set of variables (vertices), it starts from a complete graph $\mathcal{G}$. By a series of statistical tests on each pair of variables, all dispensable edges $X$ — $Y$ are removed if a (conditional) independence and separating set $C$ can be found, *i.e.* $(X \perp\!\!\!\perp Y \mid C)$ (step 1). The resulting undirected graph is called the **skeleton**. V-structures are then identified, $X \to Z \leftarrow Y$, if $(X \perp\!\!\!\perp Y \mid C)$ and $Z \notin C$ (step 2). Additional assumptions (e.g., acyclicity) allow for the propagation of v-structure orientations to some of the remaining undirected edges (Zhang, 2008) (step 3).

---

**Algorithm 1** The PC Algorithm

---

**Require:** $V, \mathcal{D}(V)$, significance level $\alpha$
    **Step 1**: Find the graph skeleton and separating sets of removed edges
    **Step 2**: Orient v-structures based on separating sets
    **Step 3**: Propagate orientations of v-structures to as many remaining undirected edges as possible
    **return** Output graph

---

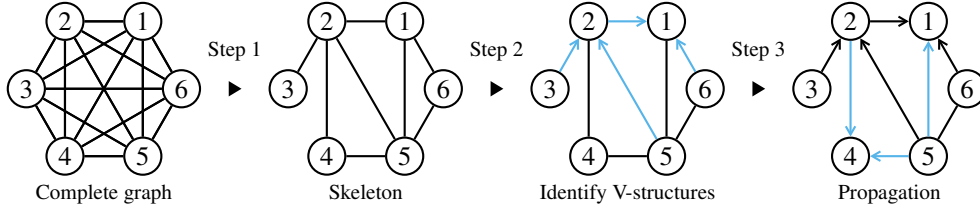

Figure 1: General procedure of constraint-based structure learning.

While the oracle version of the PC-algorithm has been shown to be sound and complete, its application is known to be sensitive to the finite size of real life datasets. In particular, the PC-algorithm in its original implementation (Spirtes, Glymour, and Scheines, 2000) is known to be order-dependent, in the sense that the output depends on the lexicographic order of the variables. This issue can be circumvented, however, for the first step of algorithm 1 with a simple modification given in algorithm 2 and referred to as Step 1 of PC-stable algorithm (Colombo and Maathuis, 2014).

---

**Algorithm 2** Find skeleton and separating sets    (Step 1 of PC-stable algorithm)

---

**Require:** Conditional independence assessment between all variables $V$ with significance level $\alpha$
    $\mathcal{G} \leftarrow \mathcal{G}_c$
    $\ell \leftarrow -1$
    **repeat**
        $\ell \leftarrow \ell + 1$
        **for** all vertices $X_i \in \mathcal{G}$ **do**
        **end for**
        $a(X_i) = \mathrm{adj}(\mathcal{G}, X_i)$
        **repeat**
            select a new pair of vertices $(X_i, X_j)$ adjacent in $\mathcal{G}$ and satisfying $|a(X_i) \backslash \{X_j\}| \geq \ell$
            **repeat**
                choose new $C \subseteq a(X_i) \backslash \{X_j\}, |C| = \ell$
                **if** $(X_i \perp\!\!\!\perp X_j | C)_\alpha$ **then**
                    Delete edge $X_i$ — $X_j$ from $\mathcal{G}$
                    $\mathrm{Sepset}(X_i, X_j \mid \mathcal{G}) = \mathrm{Sepset}(X_j, X_i \mid \mathcal{G}) \leftarrow C$
                **end if**
            **until** $X_i$ and $X_j$ are no longer adjacent in $\mathcal{G}$ or all $C \subseteq a(X_i) \backslash \{X_j\}$ with $|C| = \ell$ have been considered
        **until** all pairs of adjacent vertices $(X_i, X_j)$ in $\mathcal{G}$ with $|a(X_i) \backslash \{X_j\}| \geq \ell$ have been considered
    **until** all pairs of adjacent vertices $(X_i, X_j)$ in $\mathcal{G}$ satisfy $|a(X_i) \backslash \{X_j\}| \leq \ell$
    **return** $\mathcal{G}$, sepsets

---

## 2.2 The Consistent PC Algorithm

### 2.2.1 Lack of Robustness and Consistency of Constraint-based Methods

Beyond the order-dependence of the PC Algorithm, the general lack of robustness of constraint-based methods stems from their tendency to uncover spurious conditional independences (false negatives) between variables. This trend originates from the fact that conditioning on other variables amounts to "slicing" the available data into smaller and smaller subsets, corresponding to different combinations of categories or discrete values of the conditioning variables, over which independence tests are essentially "averaged" to assess conditional independence.

Hence, by making sure that all separating sets are actually consistent with the final graph, one expects to reduce the number of false negative edges due to spurious conditional independences inferred during the edge pruning process and, thereby, to improve the sensitivity (or recall) of the PC or PC-stable algorithms.

The inconsistency of separating sets can be of different forms, regarding either the skeleton (type I) or the final (partially) oriented graph (type II), as illustrated on Figure 1.

A type I inconsistency corresponds to a conditional independence relation such as $(2 \perp\!\!\!\perp 6 \mid 3)$ in Figure 1, for which there is no path between vertex 2 and 6 that passes through 3. This type of inconsistency often involves edges evaluated early on in the pruning process when few edges have been removed, and thus the combinatorial space of possible separating sets is still large. In particular, edge $3 — 6$, which is eventually removed in the final graph, may still exist when the edge $2 — 6$ is under consideration.

A type II inconsistency is a different kind of incompatibility originating from the orientation of the skeleton. It occurs, in particular, when a conditional independence relation is conditioned on at least one common descendant of the pair of interest in the final graph, *e.g.* $(3 \perp\!\!\!\perp 6 \mid 1)$ in Figure 1. Since it stems from the orientation of edges (steps 2&3), the origin of type II inconsistencies is generally more complex and results from a cascade of errors in both conditional independence tests and orientation.

These two types of inconsistency help define the following consistent set for candidate nodes of separating sets in absence of latent variables:

**Definition 1** (Consistent set). Given a graph $\mathcal{G}(\boldsymbol{V}, \boldsymbol{E})$ and a set of variables $\{X, Y, Z\} \subseteq \boldsymbol{V}$,

$$\mathrm{Consist}(X, Y \mid \mathcal{G}) = \{ Z \in \mathrm{adj}(X) \setminus \{Y\} \mid 1. \text{ at least one path } \gamma_{XY}^{Z} \text{ exists in } \mathcal{G};$$
$$2.\ Z \text{ is not a child of } X \text{ in } \mathcal{G} \}$$

where $\gamma_{XY}^{Z}$ is a path from $X$ to $Y$ passing through $Z$. Note that for an undirected graph, the second condition is always satisfied.

### 2.2.2 Consistent PC Pseudocodes

**Definition 2.** $\mathrm{NewStep1}(\mathcal{G}_1|\mathcal{G}_2)$ is a modified version of PC-stable step 1 (algorithm 2) where,

1. $\mathcal{G}_c$ is replaced by $\mathcal{G}_1$, and

2. $a(X_i) \setminus \{X_j\}$ is replaced by $a(X_i) \setminus \{X_j\} \cap \mathrm{Consist}(X_i, X_j \mid \mathcal{G}_2)$

Note that algorithm $\mathrm{NewStep1}(\mathcal{G}_c|\mathcal{G}_c)$ corresponds to the unmodified step 1 of original PC-stable algorithm 2. By constrast, algorithm $\mathrm{NewStep1}(\mathcal{G}_c|\mathcal{G}_\emptyset)$ removes all edges corresponding to independence without conditioning, as no separating set is involved. This unconditional independence search will be noted **step 1a**, while the subsequent conditional independence search will be referred to as **step 1b**, thereafter.

**Definition 3.** $S(\mathcal{G}_1|\mathcal{G}_2)$ is a modified version of the PC-stable algorithm, where step 1 in algorithm 1 is replaced by $\mathrm{NewStep1}(\mathcal{G}_1|\mathcal{G}_2)$ from definition 2.

Then, definition 3 allows to define algorithm 3, which ensures a consistent constraint-based algorithm through an iterative call of $S$ algorithms, $(S_k)_{k \in \mathbb{N}^\star}$, following an initial **step 1a**, $\mathrm{NewStep1}(\mathcal{G}_c|\mathcal{G}_\emptyset)$. As illustrated on Figure 2 and proved below, algorithm 3 achieves separating set consistency by repeating **step 1b** and **step 2&3**, iteratively, while searching for separating sets that are consistent with the graph obtained at the previous iteration, until a limit cycle of successive graphs is reached.

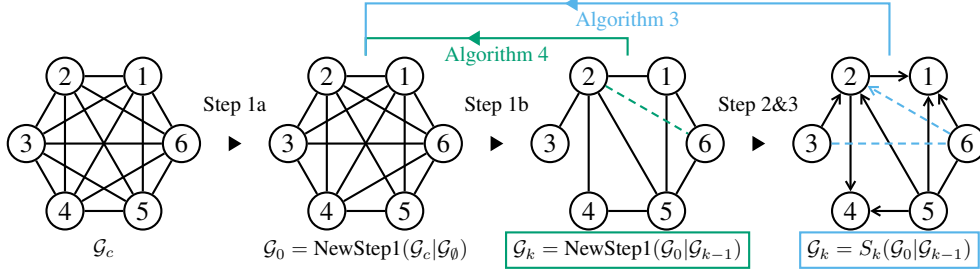

Figure 2: Illustration of the iterative procedure to learn graphical models with orientation-consistent (algorithm 3) or skeleton-consistent (algorithm 4) separating sets. Dashed edges mark the difference between two successive iterations. Proof of separating set consistency is given in theorem 4.

---

**Algorithm 3** Sepset consistent PC algorithm (1st version, orientation consistency)

---

**Require:** $V, \mathcal{D}(V)$, significance level $\alpha$
**Ensure:** $\mathcal{G}$ with consistent separating sets
   $\mathcal{G}_0 \leftarrow \text{NewStep1}(\mathcal{G}_c|\mathcal{G}_\emptyset)$
   $k \leftarrow 0$
   **repeat**
      $k \leftarrow k + 1$
      $\mathcal{G}_k \leftarrow S_k(\mathcal{G}_0|\mathcal{G}_{k-1})$
   **until** loop detected, *i.e.*, $\exists n > 0, \mathcal{G}_{k-n} = \mathcal{G}_k$
   $\mathcal{G} \leftarrow \bigcup (\mathcal{G}_j)_{j=k-n}^{k}$, with discarded conflicting orientations
   **return** $\mathcal{G}$ and consistent separating sets

---

Alternatively, one may require a separating set consistency at the level of the skeleton only, *i.e.*, before the orientation steps, which corresponds to algorithm 4, below. Indeed, early sepset inconsistencies at the level of the skeleton might cause orientation errors, which in turn can lead to the rejection of valid consistent separating sets in algorithm 3. As outlined in Figure 2, the modification of algorithm 4 only concerns step 1b, which is called iteratively until a limit cycle is reached. Then, the orientation steps 2&3 are performed as for classical PC or PC-derived algorithms, but using consistent separating sets with respect to the union of skeletons returned by the iterative call of step 1b in algorithm 4. However, as the orientation steps 2&3 might induce additional type II inconsistencies, algorithm 4 requires a final consistency check for all separating sets with respect to the final graph $\mathcal{G}$.

---

**Algorithm 4** Sepset consistent PC algorithm (2nd version, skeleton consistency)

---

**Require:** $V, \mathcal{D}(V)$, significance level $\alpha$
**Ensure:** $\mathcal{G}$ with consistent separating sets
   $\mathcal{G}_0 \leftarrow \text{NewStep1}(\mathcal{G}_c|\mathcal{G}_\emptyset)$
   $k \leftarrow 0$
   **repeat**
      $k \leftarrow k + 1$
      $\mathcal{G}_k \leftarrow \text{NewStep1}(\mathcal{G}_0|\mathcal{G}_{k-1})$
   **until** loop detected, *i.e.*, $\exists n > 0, \mathcal{G}_{k-n} = \mathcal{G}_k$
   $\mathcal{G} \leftarrow \bigcup (\mathcal{G}_j)_{j=k-n}^{k}$ and consistent separating sets with respect to the graph skeleton $\mathcal{G}$
   **Step 2** (orientation of v-structures in $\mathcal{G}$)
   **Step 3** (propagation of orientations in $\mathcal{G}$)
   **for all** removed edges $(X, Y)$ in $\mathcal{G}$ **do**
      $\text{Sepset}(X, Y \mid \mathcal{G}) \leftarrow \text{Sepset}(X, Y \mid \mathcal{G}_k)$
      **if** $\text{Sepset}(X, Y \mid \mathcal{G}) \not\subseteq \text{Consist}(X, Y \mid \mathcal{G})$ and $\text{Sepset}(X, Y \mid \mathcal{G}) \not\subseteq \text{Consist}(Y, X \mid \mathcal{G})$ **then**
         Add undirected edge $(X, Y)$ to $\mathcal{G}$
      **end if**
   **end for**
   **return** $\mathcal{G}$ and consistent separating sets

---

**Theorem 4.** *The separating sets returned by algorithms 3 and 4 are consistent with respect to the final graph $\mathcal{G}$.*

*Proof.* Firstly, the limit cycles in algorithms 3 and 4 are warranted to be finite by the deterministic nature of these algorithms and the finite set of graphs $\mathcal{G}_j$.

In algorithm 3, as the union of graphs $\bigcup (\mathcal{G}_j)_{j=k-n}^{k}$ does not remove any edge from the last graph $\mathcal{G}_k$ and discards all conflicting orientations with previous graphs $\mathcal{G}_j$, $j \in \{ k-n, k-1 \}$, taking the union of graphs does not create any new conditional independence relation, nor any inconsistency regarding the final separating sets. More precisely, all removed edges in $\mathcal{G}_k$ have separating sets consistent with respect to at least one graph in the union ($\mathcal{G}_{k-1}$), which is thus also consistent with respect to the union of graphs $\mathcal{G}$.

In algorithm 4, the consistency of separating sets is guaranteed by similar arguments, but only with respect to the skeleton. As the orientation and propagation steps 2&3 might induce additional type II inconsistencies, algorithm 4 requires a final consistency check for all separating sets. Adding back edges with inconsistent separating sets in the final graph $\mathcal{G}$ then guarantees that all the separating sets are consistent with respect to definition 1. □

### 2.2.3 Tests of Consistency

A unitary operation of algorithms 3 and 4 is to test, for a vertex $Z \in \text{adj}(X) \setminus \{ Y \}$ in $\mathcal{G}$, if $Z \in \text{Consist}(X, Y \mid \mathcal{G})$, which requires that 1) at least one path from $X$ to $Y$ passing through $Z$ (*i.e.* $\gamma_{XY}^{Z}$) exists in $\mathcal{G}$ and 2) $Z$ is not a child of $X$ in $\mathcal{G}$ (definition 1).

To test the first condition, it is conceptually simple to first get all paths between $X$ and $Y$, then check if $Z$ lies in at least one of them, This is however unfeasible as the complexity of getting all paths between two vertices can be large, depending on the edge density of the graph. Fortunately, it is possible to get directly the set of all $Z$ for which at least on path $\gamma_{XY}^{Z}$ exists. This can be done very efficiently with the help of biconnected component analysis based on block-cut tree decomposition, as detailed in Supplementary Material.

The second condition assumes the absence of latent variables, which allows for condition independence tests on adjacent nodes only in algorithm 2. It is thus straightforward to test without additional complexity burden.

Hence, the overall complexity of the consistency tests of separating sets relies on the block-cut tree decomposition, which can be done beforehand within a single depth first search with complexity $\mathcal{O}(|\boldsymbol{V}| + |\boldsymbol{E}|)$. Thus for each pair $(X, Y)$, the complexity of finding all candidate $Z$ depends on the size of the block-cut tree, which is in the worst case (when the underlying skeleton is a forest) linear in the size of the graph, $\mathcal{O}(|\boldsymbol{V}| + |\boldsymbol{E}|)$, see Supplementary Material.

### 2.3 Empirical Evaluation

We conducted a series of benchmark structure learning simulations to study the differences between the original PC-stable algorithm and the proposed modifications ensuring consistent separating sets.

For each simulation setting, we first quantified the fraction of inconsistent separating sets predicted by the original PC-stable algorithm, Figure 3. We then compared the performance of the original PC-stable (algorithm 1 and algorithm 2), orientation-consistent PC-stable (algorithm 3) and skeleton-consistent PC-stable (algorithm 4), for different significance levels $\alpha$, in terms of the precision and recall of the adjacencies found in the inferred graph with respect to the true skeleton, Figures 4 and 5. Figure 4 highlights situations for which the original PC manages to recover a DAG that is already closely related to the ground truth but produces inconsistent separating sets, as shown in Figure 3. By constrast, Figure 5 highlights standard benchmarks from the BNlearn repository (Scutari, 2010) for which the original PC show a poor Recall due to too many spurious conditional independences, and ultimately outputs a graph with only a few obvious edges. Finally, we also measured the fraction of the separating sets used for discarding edges by the three approaches that correspond to true D-separation in the ground-truth DAG, Figure 6.

### 2.3.1 Data generation and benchmarks

The data-sets used for the numerical experiments were generated with the following scheme. The underlying DAGs were generated with TETRAD (Scheines et al., 1998) as scale-free DAGs with 50 nodes ($\alpha = 0.05$, $\beta = 0.4$, average total degree $d(G) = 1.6$) using a preferential attachment model and orienting its edges based on a random topological ordering of the vertices. Data-sets were simulated with linear structural equation models for three settings : strong, medium and weak interactions (with respective coefficient ranges $[0.2, 0.7]$, $[0.1, 0.5]$, and $[0, 0.3]$ and covariance ranges $[0.5, 1.5]$, $[0.5, 1]$, and $[0.2, 0.7]$). In addition, we also generated data-sets for the classical benchmarks Insurance (27 nodes, 52 links, 984 paramaters), Hepar2 (70 nodes, 123 links, 1453 paramaters) and Barley (48 nodes, 84 links, 114005 paramaters) networks from the Bayesian Network repository (Scutari, 2010).

Reconstruction benchmarks were performed with pcalg's (Kalisch et al., 2012) PC-stable implementation, modified for enforcing separating set consistency either taking into account orientations (algorithm 3) or at the level of the skeleton (algorithm 4). The (conditional) independence test used in all simulations is a linear (partial) correlation with Fisher's z-transformation. Performances are obtained with relation to the true skeleton by measuring the Precision (positive predictive value), $Prec = TP/(TP + FP)$ and Recall or Sensitivity (true positive rate), $Rec = TP/(TP + FN)$ where $TP$ is a correctly predicted adjacency, $FP$ an incorrectly predicted adjacency and $FN$ an incorrectly discarded adjacency.

### 2.3.2 Benchmark Results

The fraction of inconsistent separating sets that were used to remove edges was first estimated for increasing sample size and varying parent-child interaction strength, using the original PC-stable algorithm for random and scale-free DAGs of 50 nodes, Figure 3. We note that in typical settings, a significant fraction of the separating sets that were used to remove edges during Step 1 of the PC-stable algorithm cannot be "read off" the returned graph, either because there is no path containing $Z$ that connects $X$ and $Y$ (skeleton inconsistency, green in Figure 3) or because there is a conditioning on an invalid child node (orientation inconsistency, *i.e.*, difference between blue and green inconsistencies in Figure 3). Both increasing the sample size and increasing the interaction strength reduces the number of inconsistent sepsets. We attribute this in part to the severity of the PC-stable algorithm which tends to remove to many false negative edges because of spurious inconsistencies. With a larger sample size $N$ and stronger interactions, consistent separating sets are still not guaranteed by the original algorithm but these settings decrease the number of spurious independencies and leads to denser reconstructed graphs, thus making it more likely for potential separating sets to be consistent. Orientation consistency is particularly difficult to obtain with respect to the returned CPDAG, as orientation and propagation steps generally suffer even more from sampling noise and previous mistakes than the skeleton reconstruction (Step 1). Notably, the orientation depends on the order in which separating sets are tested in PC-stable (in pcalg it depends on the ordering of the variables in the data-set).

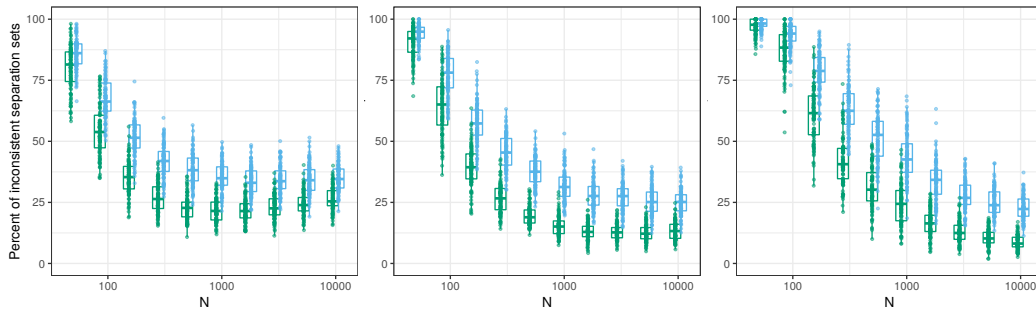

Figure 3: **Sepset inconsistency of the original PC-stable algorithm.** In each subplot the fraction of inconsistent separating sets with respect to the skeleton (green) or CPDAG (blue) obtained with the original PC-stable algorithm with a fixed $\alpha = 0.05$ is displayed for increasing sample size $N$. Data-sets were generated from 100 scale-free graphs of 50 nodes and $d(G) = 1.6$ with different parent-child interaction strengths : strong (left), medium (middle) and weak (right).

We then compared the performance of the original PC-stable (algorithm 1 and algorithm 2), orientation-consistent PC-stable (algorithm 3) and skeleton-consistent PC-stable (algorithm 4), for different significance levels $\alpha$, in terms of the precision and recall of the adjacencies found in the inferred graph with respect to the true skeleton, Figures 4, 5 and S1. Enforcing the sepset consistency is shown to significantly improve the sensitivity of constraint-based methods, for a given $\alpha$, while achieving equivalent or better overall structure learning performance.

It is particularly the case for standard benchmark networks from the BNlearn repository (Scutari, 2010), Figure 5, for which the original PC-stable algorithm shows good precision but poor recall (Rec<0.15-0.35 and Prec>0.65 at maximum Fscore, see iso-Fscore dotted lines in Figure 5), while consistent PC-stable achieves a better balance between precision and recall (Rec$\simeq$0.5 and Prec$\simeq$0.5-0.6 at maximum Fscore, Figure 5).

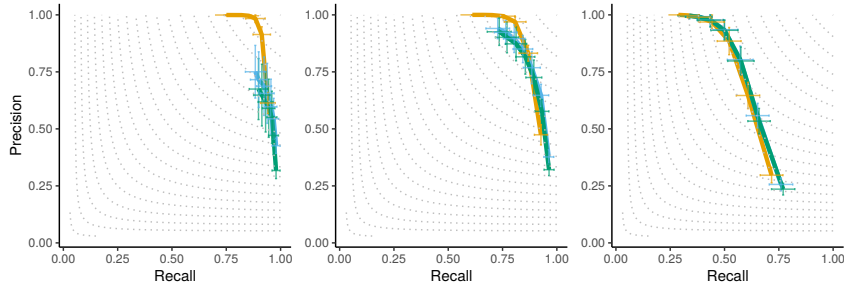

Figure 4: **Precision-recall curves for the original PC-stable (yellow), skeleton-consistent PC-stable (green) and orientation-consistent PC-stable (blue)**. The mean performances and standard deviations (error bars) obtained over 100 networks are shown for 7 values of the (conditional) independence significance threshold $\alpha$ between $10^{-5}$ and 0.2 Data-sets with $N$=500 samples were generated from the same graphs as in Figure 3 with strong (left), medium (middle) and weak (right) interactions. See Figure S1 for $N$=100, 1000.

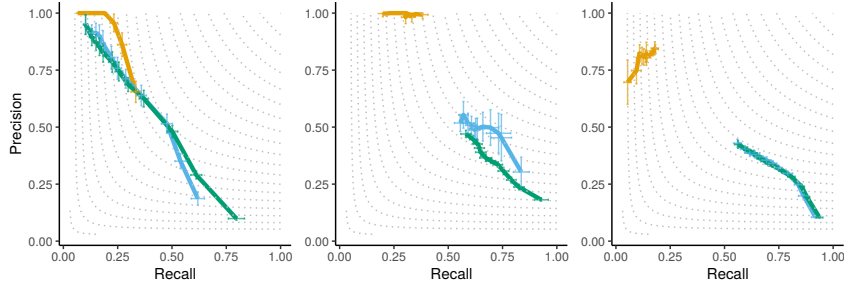

Figure 5: **Precision-recall curves for the original PC-stable (yellow), skeleton-consistent PC-stable (green) and orientation-consistent PC-stable (blue)**. The mean performances and standard deviations (error bars) obtained over 100 networks are shown for 12 values of the (conditional) independence significance threshold $\alpha$ between $10^{-25}$ and 0.5 (1e-25 1e-20 1e-17 1.0e-15 1.0e-13 1.0e-10 8.7e-09 7.6e-07 6.6e-05 5.7e-03 5.0e-02 5.0e-01). Data-sets with $N$=1000 samples were generated for the standard benchmarks Hepar2 (left), Insurance (middle) and Barley (right) networks from the BNlearn repository (Scutari, 2010).

Finally, we also compared the fraction of valid separating sets used for discarding edges, which entail true d-separation in the ground-truth DAG, Figures 6 and S2. Ensuring the consistency of separating sets tends to increase, although not guarantee, their validity in terms of actual d-separation. Consistent sepsets with invalid d-separation are primarily caused by edge mis-orientations rather than skeleton errors. In particular, skeleton-consistent separating sets yield better performance in terms of valid d-separation than orientation-consistent separating sets with the setting of the PC-stable algorithm used here. This is, however, expected to depend on the specific settings for conditional independence test, orientation and propagation rules, used in different constraint-based methods.

## 3  Conclusion

In this paper, we propose and implement simple modifications of the PC algorithm also applicable to any PC-derived constraint-based methods, in order to enforce the consistency of the separating sets

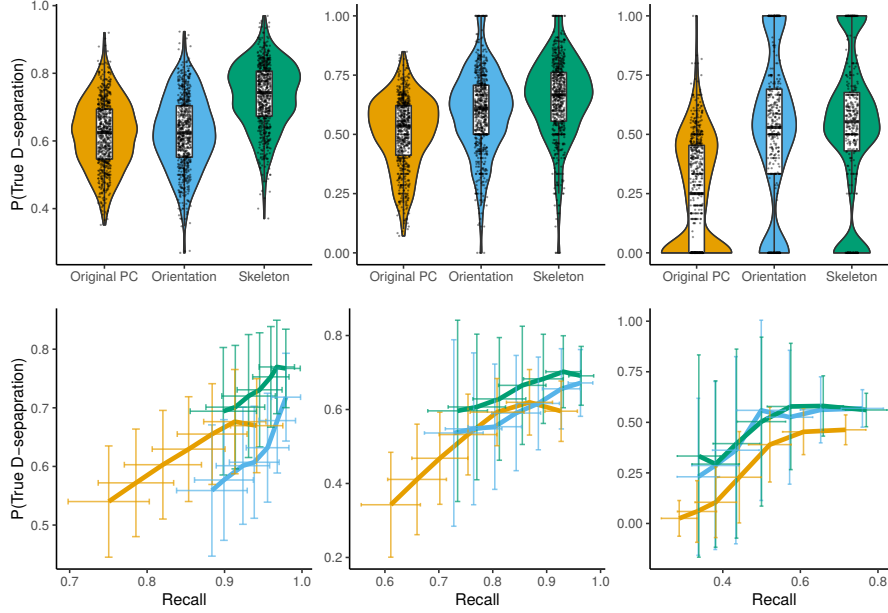

Figure 6: **Proportion of valid d-separation sepsets among edge-removing sepsets.** Top row shows the proportion of sepsets that correspond to a valid d-separation in the true DAG that were used for removing edges during Step 1 of original, orientation-consistent and skeleton-consistent PC-stable algorithms for all tested $\alpha$. Bottom row shows the average proportion of valid d-separation for a given average recall over all tested values of $\alpha$. Data-sets with $N$=500 were generated from 100 DAGs with linear SEMs with strong (left), medium (middle) and weak (right) interaction (see Figure S2 for $N$=100, 1000).

of discarded edges with respect to the final graph, which is an actual shortcoming of constraint-based approaches, Figure 3. Enforcing sepset consistency is shown to significantly improve the sensitivity of constraint-based methods, while achieving equivalent or better overall structure learning performance, Figures 4, 5 and S1. In addition, ensuring the consistency of separating sets increases also their validity in terms of actual d-separation, Figures 6 and S2.

The existence of sepset inconsistencies with constraint-based methods originates from their tendency to uncover spurious conditional independences early on in the pruning process when the combinatorial space of possible separating sets is still large, unlike in the final typically sparse skeleton. Such spurious conditional independences are responsible, in particular, for the large number of false negative edges and, therefore, frequently poor sensitivity of constraint-based methods (Colombo and Maathuis, 2014). By contrast, enforcing sepset consistency enables to achieve a better balance between sensitivity and precision.

To circumvent this inconsistency issue during the skeleton step, we have shown that one can either use sepset consistency taking into account orientations to help reject inconsistent sepsets (algorithm 3) or use sepset consistency of the skeleton to help determine the orientations (algorithm 4). The later approach tends to yield slightly better performance with the setting of the PC-stable algorithm used here but this is expected to be dependent on the specific settings used, for conditional independence test, orientation and propagation rules, in different constraint-based methods.

Indeed, the methods and algorithmic implementations presented here are not primarily meant to out-compete a specific PC or PC-derived algorithm but rather to improve the explainability of constraint-based methods, by ensuring the consistency of all separating sets in the final causal graphs.

The approach is very general and applicable to the large variety of constraint-based methods, starting with a complete graph and discarding dispensable edges iteratively based on conditional independence search. Beyond the formal interest of guaranteeing sepset consistency, this is also especially important, in practice, for the interpretability of constraint-based models for real-life applications.

### Acknowledgements
The authors acknowledge financial support from the French Ministry of Higher Education and Research, PSL Research University and Sorbonne University.

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
