[Supplementary Material]

<div align="center">

# SUPPLEMENTARY MATERIAL

*on NIPS 2019 paper*

## Constraint-based Causal Structure Learning with Consistent Separating Sets

</div>

<div align="center">

Honghao Li,   Vincent Cabeli,   Nadir Sella,   Hervé Isambert
Institut Curie, PSL Research University, CNRS UMR168, Paris
{honghao.li, vincent.cabeli, nadir.sella, herve.isambert}@curie.fr

</div>

An R implementation of the methods in the case of the PC-stable algorithm is available with examples at https://github.com/honghaoli42/consistent_pcalg.

## A   Test of Consistency

### A.1   Terminology

A **connected graph** $\mathcal{G}$ is such that there is a path between each pair of vertices of $\mathcal{G}$. A **connected component** of a graph is a maximal connected subgraph. An **articulation point** (or **cut point**) is a vertex in a connected graph whose removal would disconnect the graph and thus increase its number of connected components. A **biconnected graph** is a connected graph without articulation point. A **biconnected component** (or **block**) is a maximal biconnected subgraph.

### A.2   Biconnected Component Analysis

For a pair $(X, Y)$ in a graph $\mathcal{G}$, one of the necessary conditions for its separating set to be consistent, as stated in definition 1, is that for each vertex $Z$ in the separating set, $Z$ lies on a path $\gamma_{XY}^Z$ between $X$ and $Y$ in the skeleton of $\mathcal{G}$. For one pair of vertices, checking the existence of a path for all $Z$ can already be time consuming if the degrees of the vertices are large. In addition, the complexity will be further multiplied by the number of pairs to be considered. Fortunately, it is possible to avoid this high complexity with the help of the biconnected component analysis based on block-cut tree decomposition, and thus to limit the search of consistent separating vertices within those that are consistent with respect to the skeleton.

**Definition 5** (Block-cut tree). $\mathcal{G}$ a connected (sub)graph. The block-cut tree decomposition of $\mathcal{G}$ is denoted by $\mathcal{T}(\boldsymbol{B}, \boldsymbol{C}, \boldsymbol{Br})$ where $\boldsymbol{B} = \{ b_i \}_{i=1}^m$ is the set of biconnected components (or blocks) of $\mathcal{G}$, $\boldsymbol{C} = \{ c_j \}_{j=1}^n$ is the set of articulation points (or cut points) and $\boldsymbol{Br} = \{ (b_i, c_j) \mid b_i \in \boldsymbol{B}, c_j \in \boldsymbol{C}, b_i$ and $c_j$ are adjacent in $\mathcal{T} \}$ is the set of connections between $\boldsymbol{B}$ and $\boldsymbol{C}$.

In the following we establish a relation between biconnected components and the path existence problem.

**Lemma 6** (Menger's theorem for biconnected graph). *Let $\mathcal{G}(\boldsymbol{V}, \boldsymbol{E})$ be a biconnected graph, $\{ X, Y \} \subseteq \boldsymbol{V}$ a pair of vertices. There is a cycle in $\mathcal{G}$ that contains $X$ and $Y$.*

**Theorem 7.** *Let $\mathcal{G}(\boldsymbol{V}, \boldsymbol{E})$ be an undirected graph, $\mathcal{H}(\boldsymbol{V_\mathcal{H}}, \boldsymbol{E_\mathcal{H}}) \subseteq \mathcal{G}$ a biconnected component of $\mathcal{G}$, $\{ X, Y \} \subseteq \boldsymbol{V_\mathcal{H}}$ a pair of vertices, and $Z \in \boldsymbol{V_\mathcal{G}}$ a third vertex. There is a path $\gamma_{XY}^Z$ if and only if $Z \in \boldsymbol{V_\mathcal{H}}$.*

*Proof.* If there is a path $\gamma_{XY}^Z$, suppose that $Z \notin \boldsymbol{V_\mathcal{H}}$, then the subgraph $\mathcal{H}'$ of $\mathcal{G}$ over $\boldsymbol{V_\mathcal{H}} \cup \{ Z \}$ is biconnected thanks to $\gamma_{XY}^Z$, and $\mathcal{H} \subset \mathcal{H}'$ is not a biconnected component of $\mathcal{G}$ as it is not maximal. Therefore we must have $Z \in \boldsymbol{V_\mathcal{H}}$.

If $\{ X, Y, Z \} \subseteq \boldsymbol{V_\mathcal{H}}$, then lemma 6 guarantees a cycle that contains $Z$ and $Y$. Since $\boldsymbol{V_\mathcal{H}}$ contains at least three vertices, such a cycle contains $n \geq 1$ vertices other than $Z$ and $Y$, and can be represented by two edge-distinct paths between $Z$ and $Y$:

$$\gamma_{ZY}^{(1)} = ZU_1 U_2 \cdots U_k Y, \quad \gamma_{ZY}^{(2)} = ZU_{k+1}U_{k+2} \cdots U_n Y$$

where $k \in \mathbb{Z}^{\geq 0}$ (with $k = 0$ indicating a direct edge between $Z$ and $Y$), $n \in \mathbb{Z}^+$, $k < n$ and $\{ U_i \}_{i=1}^n$ are distinct vertices. Since $Y$ is not an articulation point, there is a path $\gamma_{XZ}$ that does not contain $Y$:

$$\gamma_{XZ} = X D_1 D_2 \cdots D_m Z$$

where $m \in \mathbb{Z}^{\geq 0}$ and $\{ D_j \}_{j=1}^{m}$ are distinct vertices. If $\{ U_i \}_{i=1}^{n} \cap \{ D_j \}_{j=1}^{m} = \emptyset$, then there is a path

$$\gamma_{XY}^{Z} = \gamma_{XZ}\gamma_{ZY}^{(i)}, i \in \{ 1, 2 \}.$$

Otherwise, suppose $\{ U_i \}_{i=1}^{n} \cap \{ D_j \}_{j=1}^{m} = \{ D_{p_1}, D_{p_2}, \ldots, D_{p_t} \}$ where $t \in \mathbb{Z}^{+}$ and $p_1 < p_2 < \cdots < p_t$, and suppose $D_{p_1} = U_l$. If $l \leq k$, then there is a path

$$\gamma_{XY}^{Z} = XD_1D_2 \cdots D_{p_1}(U_l)U_{l-1} \cdots U_1\gamma_{ZY}^{(2)},$$

if $l > k$, then there is a path

$$\gamma_{XY}^{Z} = XD_1D_2 \cdots D_{p_1}(U_l)U_{l-1} \cdots U_{k+1}\gamma_{ZY}^{(1)}.$$

As a result, if $\{ X, Y, Z \} \subseteq \boldsymbol{V}_{\mathcal{H}}$, then there is always a path $\gamma_{XY}^{Z}$. $\qquad\square$

**Corollary 8.** *Let $\mathcal{G}(\boldsymbol{V}, \boldsymbol{E})$ be a connected graph, $\mathcal{T}(\boldsymbol{B}, \boldsymbol{C}, \boldsymbol{Br})$ the block-cut tree decomposition of $\mathcal{G}$, $\{ X, Y \} \subseteq \boldsymbol{V}$ a pair of vertices, $n_X, n_Y$ the corresponding nodes of $X$ and $Y$ in $\mathcal{T}$, and $\boldsymbol{S} = \{ Z \in \boldsymbol{V} \setminus \{ X, Y \} \mid$ at least one path $\gamma_{XY}^{Z}$ exists. $\}$*

1. *If $n_X = n_Y = b_i \in \boldsymbol{B}$, then $\boldsymbol{S} = \boldsymbol{V}(b_i) \setminus \{ X, Y \}$.*

2. *If $n_X \neq n_Y$, let $\nu_{XY} = w_1 w_2 \cdots w_k, w_1 = n_X, w_k = n_Y$ be the path between $n_X$ and $n_Y$ where each $w_i$ belongs to $\boldsymbol{B}$ or $\boldsymbol{C}$, then $\boldsymbol{S} = \bigcup (\boldsymbol{V}(w_i))_{i=1}^{k} \setminus \{ X, Y \}$.*

The first case is a direct result of theorem 7. The second case is not difficult to prove once we notice the fact that $\nu_{XY}$ is the unique path between $n_X$ and $n_Y$ in $\mathcal{T}$, and that every $\gamma_{XY}$ must contain all the cut points in $\nu_{XY}$, and thus can be decomposed into segments of paths between these cut points.

Each undirected graph $\mathcal{G}(\boldsymbol{V}, \boldsymbol{E})$ can be decomposed into a set of single vertices and a set of connected subgraphs, where each subgraph can be represented by a block-cut tree. Based on this decomposition, algorithm 5 gives the consistent candidate vertices for separating set for a pair of vertices as described in definition 1.

---

**Algorithm 5** Consistent candidates

---

**Require:** (Partially directed) graph $\mathcal{G}(\boldsymbol{V}, \boldsymbol{E})$, its block-cut tree decomposition for each connected component (with respect to its skeleton) $\{ \mathcal{T}_i(\boldsymbol{B}, \boldsymbol{C}, \boldsymbol{Br}) \}$, two vertices $\{ X, Y \} \subseteq \boldsymbol{V}$
**Ensure:** Set of all candidate vertices $\text{Consist}(X, Y \mid \mathcal{G})$.
    **if** $X$ and $Y$ do not belong to the same block-cut tree $\mathcal{T}_i$ **then**
        **return** $\emptyset$
    **end if**
    **if** $X$ and $Y$ belong to the same block $b_i \in \boldsymbol{B}$ **then**
        **return** $(\text{Ne}(X) \setminus \text{Child}(X)) \cap (\boldsymbol{V}(b_i) \setminus \{ X, Y \})$
    **else**
        $\nu_{XY} \leftarrow \text{TreePath}(n_X, n_Y) = w_1 w_2 \cdots w_k$
        **return** $(\text{Ne}(X) \setminus \text{Child}(X)) \cap (\bigcup (\boldsymbol{V}(w_i))_{i=1}^{k} \setminus \{ X, Y \})$
    **end if**

---

The block-cut tree decomposition can be done beforehand within a single depth first search with complexity $\mathcal{O}(|\boldsymbol{V}| + |\boldsymbol{E}|)$. Thus for each pair $(X, Y)$, the complexity of finding all candidate $Z$ depends on the size of the block-cut tree. In the worst case where $\mathcal{G}$ is a forest with only bridges (edges, the removal of each bridge increases the number of connected components of $\mathcal{G}$), the number of nodes and branches in the block-cut tree $\mathcal{T}$ of $\mathcal{G}$ is of the same order of $|\boldsymbol{V}|$ and $|\boldsymbol{E}|$, and for all pair of vertices $\{ X, Y \} \subseteq \boldsymbol{V}$ we need to perform a path search in $\mathcal{T}$ of complexity $\mathcal{O}(|\boldsymbol{V}| + |\boldsymbol{E}|)$ to get $\boldsymbol{S}$. In the best scenario where $\mathcal{G}$ is biconnected, $\boldsymbol{S} = \boldsymbol{V} \setminus \{ X, Y \}$ for all pairs. Then, an operation of set intersection $(\text{Ne}(X) \setminus \text{Child}(X)) \cap \boldsymbol{S}$ with linear complexity $\mathcal{O}(|\text{Ne}(X)| + |\boldsymbol{S}|)$ will give the result.

# B   Supplementary Figures

Figure S1: **Precision-recall curves for the original PC-stable (yellow), orientation-consistent PC-stable (blue) and skeleton-consistent PC-stable (green)**. Data-sets of $N$=100 samples (top row) or of $N$=1000 (bottom row), with strong (left), medium (middle) and weak (right) interactions. See Figure 4 for more information.

Figure S2: **Proportion of valid d-separation sepsets among edge-removing sepsets found during reconstruction.** Data-sets of $N$=100 samples (top two rows) or of $N$=1000 (bottom two rows), with strong (left), medium (middle) and weak (right) interactions. See Figure 6 for more information.