[Reviews · NeurIPS 2019]

Reviewer 1



Summary ------- The authors address a major drawback of constraint-based causal structure learning algorithms (PC algorithm and derivatives), namely, that for finite sample sizes the outputted graphs may be inconsistent regarding separating sets: The final graph may imply different separating sets than those identified in the algorithm. This implies in particular that outputted graphs are not guarenteed to belong to their presumed class of graphical models, for example CPDAGs or PAGs. The main reason is that PC-based methods remove too many true links in the skeleton phase. The authors' solution is based on an iterative application of a modified version of PC until the final graph is consistent with the separating sets. They prove that their solution fixes the problem and demonstrate these improvements with numerical experiments. Strengths --------- - Improves on an important current shortcoming of the PC algorithm - theoretical results underpinning their solution Weaknesses ---------- - a bit heavy on constraint-based causal discovery jargon, some explanation for non-experts would help - computational complexity not sufficiently discussed Quality ------- The algorithm and theory is technically sound. What I miss is a more thorough discussion of the increase in compational complexity since this may be a major impediment in adopting the algorithm. The introduction could also mention other structure learning approaches that, I believe, don't have the inconsistency issue, e.g., SAT-solver based approaches [2]. One question to clarify for non-experts is whether the presented modification is relevant also for the graph class of DAGs instead of CPDAGs (no Markov ambiguity)? I.e., if only the skeleton phase is needed since all links are oriented due to time-order or other side knowledge. Also, since this is about finite sample consistency, the authors could discuss how the new algorithm relates to the uniform consistency proof of the PC algorithm [1]? Clarity ------- Overall the article well-written, but a bit heavy on constraint-based causal discovery jargon. I have listed a number of points below that could be immproved upon. What would help is a figure and intuitive example of sepset inconsistency. I find a paper much more readable that illustrates a problem and solution on a simple example. Originality ----------- As far as I know the work is new and related work is cited. Significance ------------ The new algorithm can be of significant impact since, as the authors also mention, it makes PC-based methods more robust and better interpretable. The question is just whether the increase in computational complexity makes this prohibitive. That's not clear to me from the paper. Minor further points -------------------- Abstract: - explain separating set for non-experts - "uncover spurious conditional independences" --> "uncover false negatives" ? Introduction "sampling noise" --> finite sample sizes - could cite some more related work, what about SAT solver approaches to causal discovery [2]? - "invoked to remove" --> sounds odd - "achieve a better balance" --> "achieves a better balance" Section 2.2.1 first paragraph is unclear, could phrase "slicing the available data" better... Definition 1: Here I would like to see an example of a sepset inconsistency with the final that gives a non-expert an intuition Section 2.2.2 S, S', S_0, etc: all a bit confusing. Maybe a different notation would help. "(i.e. relaxed, majority or conservative rules)" --> difficult to understand for non-experts Proof of Theorem 4 "discard all" --> "discards all" "less stringent separating set consistency" --> what does this imply theoretically? The theorem 4 still holds, right? "the the edges" --> "the edges" Section 2.3 Here one could summarize the difference in complexity between PC-stable and the proposed algorithm Section 2.3.3 - "covariance ranges" --> Covariance between what? The SCM is run with the given coefficient ranges and gaussian unit variance errors, no? - Which specific alpha levels were chosen? References ---------- [1] Kalisch, Markus. 2007. “Estimating High-Dimensional Directed Acyclic Graphs with the PC-Algorithm.” The Journal of Machine Learning Research 8: 613–36. [2] Hyttinen, Antti, Patrik O Hoyer, Frederick Eberhardt, and Matti Järvisalo. 2013. “Discovering Cyclic Causal Models with Latent Variables: A General SAT-Based Procedure.” In Proceedings of the Twenty-Ninth Conference on Uncertainty in Artificial Intelligence, 301–10. ---------------------- Update after author feedback: The authors convincingly addressed my question regarding computational complexity and they have also found a better definition of orientation-consistency. I also acknowledge that the authors will explain the problem better with an example figure. With these, I would still stick to my previous score "a good submission" (7). I concur with the other reviewers evaluation that the work is more incremental, but in a very solid way.

Reviewer 2



Updated: I have read the response and thank the authors for dealing with the points raised. My overall opinion of the paper has not changed and I have not updated my score. ----------- The paper tackles the problem of lack of consistency in Bayesian network learning algorithms based on the well known PC algorithm. Specifically, the paper proposes and analyses an algorithm to ensures the conditional independences used during learning are consistent with those in the final learned graph. As the results show, this has the pleasing effect of producing final graphs that are more accurate. The paper generally presents the problem to be solved well. The introduction arguably seems to be missing a sentence or two from the very beginning explaining the big picture or general area. The background and related work given is appropriate and sufficient, though the obligatory throw-away references to search-and-score methods looks quite dated - there has been a lot of significant improvement in this field in the last decade or so. The method can generally be understood. However, a lot of the method description is being done by the algorithms given; the text would benefit from having more signposting of contributions and more focus on how it works at a conceptual level, not just the algorithmic level. The evaluation is perfectly sensible and uses a range of different networks and measures to ensure general applicability of the results. While the text states the method can be used with any PC derived algorithm, the results only show it being applied to the main PC algorithm. It would perhaps also be useful to see the results for the method on some standard benchmark problems so that cross evaluation with other techniques could be easily made.

Reviewer 3



This paper introduces a modification of the PC algorithm to guarantee that the separating subsets of variables are, in the author terms "consistent" with the underlying graph. The use of the term "consistent" is, according to this reviewer, somewhat non-standard as, in general, it is used for an estimate that has some asymptotic properties. I would say that here it means that the separating subsets are compatible with the underlying graph. Obviously, once defined, the term "consistent" can be used but I think that a broad readership may have a different expectation on the paper when reading the term "consistent" in the current title. The paper is well written and the contribution, while somewhat incremental, seems original to this reviewer. The experimental results should demonstrate that the proposed approach performs better than the original PC-algorithm. However, unless I'm misinterpreting Figure 2, that figure shows a better precision-recall trade-off for the standard PC-algorithm (yellow line) than the two proposed variants of the algorithm (skeleton-consistent -blue- and orientation-consistent -green), specially at the high-end of the precision. A good performance at that end of the precision is important in many domains.

[Author Response · NeurIPS 2019]

We would like to thank the reviewers for their insightful comments and suggestions. Due to space limitation, we only
address their main concerns below, but all the other (more minor) issues and suggestions will be properly addressed in
the final paper (ie big picture description without causal discovery jargon, intuitive sepset (in)consistency examples,
refs to SAT-solver, application to other PC-related methods, change of title to distinguish from asymptotic consistency).

• **Reply to Reviewer 1** (*"discuss computational complexity"*):
The reviewer is right in pointing out that the complexity for ensuring sepset consistency needs to be clarified. As alluded
to in the original manuscript, the complexities including *versus* excluding edge orientation needed to be addressed
separately, although both approaches lead to similar results, in terms of performance, Figs. 2&3 and new Figure below.

The skeleton-consistency can be done at worst with a linear complexity increment, $\mathcal{O}(|\boldsymbol{V}| + |\boldsymbol{E}|)$, relative to PC
algorithm*, by using the biconnected component analysis based on block cut tree decomposition detailed in Suppl. Ma-
terial. [*PC algorithm runs in exponential time in the worst case but usually in polynomial time on sparse DAGs].

By contrast, the orientation-consistency introduced in the original submission involved the search of collider-free paths,
entailing in the worst case an exponential complexity, which could be largely mitigated, in practice, by first applying the
very efficient skeleton-consistent algorithm, as discussed in the original manuscript. However, the situation is actually
better, as we recently realized that the orientation-consistency could be implemented with the same linear complexity as
the skeleton-consistency, by exploiting PC conditional independence search restricted to $\mathrm{N}(\{X, Y\})$, the neighborhood
of $X$ and $Y$, with a more appropriate definition of orientation-consistency, ie $\mathrm{Consist}(X, Y|\mathcal{G}) = \{Z \in \mathrm{N}(\{X, Y\}) \,|$
$Z$ is on a path, $\gamma_{XY}^Z$, from $X$ to $Y$ and $Z$ is not a child of $X$ or $Y\}$. The proof of sepset consistency (Theorem 4)
using this definition is essentially the same and the results are still very similar to those obtained with the (unmodified)
skeleton-consistency approach, as seen in the new Figure below. The new complexity is thus at worst linear in all cases.

• **Reply to Reviewer 2** (*"provide additional experimental evaluation on standard benchmarks"*):
Following suggestions by Reviewers 2 & 3, we performed additional evaluations on standard benchmarks from the
Bayesian Network repository, which display a clearer performance improvement over standard PC, see Figure below.

Figure 1: Precision-recall curves (1e-25 $< \alpha <$ 0.5) for the original (yellow), skeleton-consistent (green) and (new) orientation-consistent (blue) PC-stable. Hepar2 (left), Insurance (middle) and Barley (right) benchmarks from www.bnlearn.com ($N = 1000$). Original PC: Rec $<$ 0.15-0.35 and Prec $>$ 0.65 at max Fscore, see iso-Fscore dotted lines ($\alpha_{\mathrm{opt}} =$ 0.5 / 0.5 / 6e-5, respectively); Sepset-consistent PC-stable: Rec $\simeq$ 0.5 and Prec $\simeq$ 0.5-0.6 at max Fscore $\simeq$ 0.5-0.55 ($\alpha_{\mathrm{opt}} =$ 5e-3 / 8e-7(1e-17) / 1e-20, respectively).

• **Reply to Reviewer 3** (*"show a more clear improvement over the standard PC algorithm"*):
As underlined in the last paragraph of the manuscript (which will also be emphasized earlier in the final paper),
the main improvement of our method over standard PC concerns its guarantee on the consistency of the separating
sets (Theorem 4), not its performance in terms of precision-recall plot (Fig. 2). In addition, we show that ensuring
consistency of separating sets improves their validity in terms of actual d-separation (Fig. 3).

Yet, we agree with this reviewer that our choice of empirical evaluation was not optimal... as it showcased settings for
which the original PC-stable algorithm performs already very well (ie the maximum Fscore reached for an optimal $\alpha$ is
already close to the top right corner in the precision-recall plot). However, as is well known in the field, PC-related
algorithms typically exhibit poor recalls on standard benchmarks, as shown in the examples of the new Figure above,
whereas enforcing sepset consistency leads to a better balance between precision and recall at maximum Fscore.

While the "high-end of precision" can be seen as important in order to trust the discovered edges, it also corresponds to
the "low-end of recall", which usually implies that many true edges are actually dismissed. For this reason, improving
the low recall of PC-related algorithms has been a long standing yet unachieved goal in the field (see eg section 5.2 of
Colombo et al, JMLR, 15 (2014) 3741-3782). It is especially important for the interpretability of real-life applications
where one would like to discover, beyond the obvious edges, as many non-obvious edges as possible or else the
conditional independences with consistent separating sets to explain their removal. This is the motivation for our paper.

[Meta-Review · NeurIPS 2019]

A clever extension to the PC algorithm for causal structure learning aimed to address inconsistency of results in terms of separating sets between the pruning step and the final graph. The new approach is somewhat incremental, but the authors provide some new formal guarantees. Experiments are reasonable, although they could be much better (please see reviews, this is also acknowledged by the authors, as currently one may wonder about the advantages wrt PC). I also suggest the authors to motivate the novelty and how it can improve/has improved results, in particular in view of a higher computational complexity.